# Impact of Environmental Radiation on the Incidence of Cancer and Birth Defects in Regions with High Natural Radioactivity

**DOI:** 10.3390/ijerph19148643

**Published:** 2022-07-15

**Authors:** Anastasia Zlobina, Iskhak Farkhutdinov, Fernando P. Carvalho, Nanping Wang, Tatiana Korotchenko, Natalia Baranovskaya, Anvar Farkhutdinov

**Affiliations:** 1Department of Geology, Hydrometeorology and Geoecology, Faculty of Earth Sciences and Tourism, Bashkir State University, 450074 Ufa, Russia; anastasizlobina@gmail.com (A.Z.); anvarfarh@mail.ru (A.F.); 2Scientific Department, Vernadsky State Geological Museum, 125009 Moscow, Russia; i.farkhutdinov@sgm.ru; 3Instituto Superior Técnico, Campus Tecnológico e Nuclear, University of Lisbon, 2695-066 Bobadela LRS, Portugal; 4School of Geophysics and Geoinformation Technology, China University of Geosciences, Beijing 100083, China; 1996010992@cugb.edu.cn; 5Engineering School of Natural Resources, National Research Tomsk Polytechnic University, 634030 Tomsk, Russia; tatyana1003@mail.ru (T.K.); nata@tpu.ru (N.B.)

**Keywords:** radon, uranium, thorium, HBRA, environmental radioactivity, epidemiological studies, lung cancer, leukemia, nasopharyngeal carcinoma

## Abstract

Four regions of high natural radioactivity were selected to assess radionuclide levels in rocks and soils, ambient radiation doses, radon exhalation from the ground, and radon concentrations in the air. The regions have different geochemical characteristics and radioactivity levels, which modulate the radiation exposure of local populations. Combining radiometric data with data from regional health statistics on non-infectious diseases, a statistically significant positive correlation was found between radiation exposure and the incidence of cancer and birth defects. Although this is a preliminary and prospective study, the empirical evidence gathered in this paper indicated increased the incidence of some diseases in relationship with the natural radiation background. It is suggested that further research, including epidemiological studies and direct determination of radiation exposures in regions with a high natural radiation background, is needed and justified.

## 1. Introduction

Granite rocks are known to be a major environmental deposit of natural radioactive elements (NRE). The soils formed from the weathering of such rocks generally inherit high concentrations of uranium (U), thorium (Th), and radioactive potassium (^40^K). In soils, these primordial radionuclides and their radioactive decay products, such as radium (Ra), radon gas (Rn), radioactive lead (Pb), and polonium (Po) isotopes, are always present [1,2]. Some of the high natural radiation background areas are located in regions with soils formed from granite rocks [3,4,5]. The human populations living in these areas are exposed to enhanced radiation levels and the biological effects of radiation doses have been investigated over the last decades [6,7].

The data provided by international organizations, such as the United Nations Scientific Committee on the Effects of Atomic Radiation (UNSCEAR), the International Atomic Energy Agency (IAEA), the International Commission on Radiological Protection (ICRP) and by scientific reports from several disciplines, have suggested that residents of high background radiation areas (HBRAs) may suffer from various radiation-induced health effects. These effects could be related to radiation-induced genetic, genomic, and chromosomal abnormalities, which potentially are translated into increased cancer and birth defect rates [8,9,10,11,12,13,14,15,16,17,18,19,20,21,22].

The state of Kerala in India, with abundant deposits of monazite sands rich in thorium, is one example of such areas with high natural radiation. Therein were reported high incidence of genomic pathologies among the population, frequent cytogenetic aberrations, an increased incidence of Down syndrome, and idiopathic Parkinsonism [17], clubfoot, and other types of birth defects among newborns [23]. However, other studies in the same area have concluded that the confounding factors are many and radiation induced effects might be originated by other causes [24]. For other high radiation background areas, the results from different studies have been contradictory as well [4,5].

Currently, there are no unanimous views about the effects of natural radiation on human health, and some reports claim that effects on public health are not observable in regions with high natural radioactivity [25,26]. Some authors stated that exposure to slightly enhanced radioactivity levels is even beneficial and prompts developing adaptive responses to low doses (hormesis) [27,28]. If such effect exists, the linear non-threshold model (LNT) would not apply to low doses. Although for practical purposes this might be irrelevant, the domain of such effect is at the normal radiation background, i.e., <5 mSv y^−1^ (the world average human exposure is 2.4 mSv y^−1^) [6]. At this dose rate level, it was claimed that exposure might be beneficial trough adaptive response and translated in a smaller morbidity of cancer and a longer lifetime [29]. However, in the HBRAs, the effective dose rates are in the range of 5–203 mSv y^−1^, or 0.01–0.55 mSv d^−1^ [4,30]. In animal models, protracted exposure to dose rates in the range of 0.05–21 mGy d^−1^ showed that biological effects (several types of cancer) are produced. In spite of this, no guidance and no measures were developed concerning the radiation protection of human populations living in areas with high natural radioactivity [31]. Moreover, there are claims that populations inhabiting some of these areas, such as Ramsar in Iran, are adapted and display no enhanced morbidities and therefore no protection measures would be required [32]. If confirmed, this would extend the non-linearity of dose–effect response to much higher dose levels, i.e., in the range of 5–200 mSv y^−1^.

Based on a vast body of research, the World Health Organization (WHO) recently acknowledged that prolonged exposure to Rn gas through inhalation is the second cause of lung cancer in the global population, after cigarette smoke, and accounting for 3–14% of all lung cancers. The ICRP and WHO proposed a maximum tolerable concentration of radon in indoor air of 100 Bq m^−3^ (annual average), below which no significant health effects have been observed. Continued exposure to radon at 100 Bq m^−3^ corresponds to an effective dose of about 3 mSv y^−1^ through inhalation. It must be noted that, for each additional 100 Bq m^−3^, the lung cancer incidence increases 16% per year [33]. Following ICRP and WHO recommendations, many countries adopted maximum tolerable radon concentrations in indoor air [33,34,35].

Inhabitants of HBRAs are often exposed to high radon concentrations, as we will discuss below. Furthermore, the ingestion of natural radionuclides with food and water adds an internal radiation dose to the dose received through radionuclide inhalation. The external irradiation is also generally enhanced in HBRAs. Through these several exposure pathways, the total radiation exposure may give rise to effective doses well above the recommended dose limits for the safety of human beings, in particular in uranium- and thorium-rich areas, despite the natural origin of radionuclides [36,37].

Current international safety standards set up dose limits for workers and members of the public (20 and 1 mSv y^−1^, respectively) to ensure protection against the harmful effects of ionizing radiation. However, such dose limits apply only to radiation exposures *added* to the natural radiation background [34,38]. This internationally agreed approach was a logical way to deal with the existence of variable radiation background values around the planet and still enabling the control of the impact of human activities on additional radiation exposures [34,38,39]. Such approach allowed for introducing harmonized safety standards on the *added* radiation exposures from practices and related responsibility issues, which was a significant progress in radiation protection. However, it is legitimate to ask whether the unmodified natural high radiation background can be considered as not harmful (and even healthy, as some claim) because it is *natural*. In other words, is the sievert (Sv) from natural high radiation background less harmful to public health than the sievert added by human activities? The recent recommendations made by ICRP and WHO regarding radon exposure in dwellings and workplaces already provided a partial answer to this question.

Regarding the HBRAs, where ionizing radiation doses up to 203 mSv y^−1^ have been reported, no common understanding has been reached. Eventually, no sufficient attention was paid to this issue yet and no sufficient research or well-designed research has been carried out in the areas with high natural radiation to fully elucidate the radiation exposure (total effective dose) and the radiation effects on human health. Hendry at al. reviewed the limitations of previous studies and difficulties in finding biological effects of radiation [4]. These authors pointed out that one key limiting factor of studies is the small size of the human population living in the areas investigated. For example, in Ramsar, Iran, the number of most exposed people would be about 100 people. It must be added that many studies on biological effects performed in high background radiation areas (HBRAs), e.g., on chromosome aberrations in lymphocytes of peripheral blood, did not include adequate radiation characterization. Furthermore, most studies of radiation exposures and dose assessments in HBRAs were based on external radiation measurements (gamma radiation) only and did not include radon measurements [4,5,40].

As a first step to assess whether biological effects are observable in large population groups inhabiting HBRAs, a prospective study was undertaken. In this study, four regions with high natural radiation background were selected to determine key radionuclide concentrations in rocks, soils, and surface air and to assess the human exposure to ambient radiation. These results, supplemented with data available in the scientific literature and public health statistics, were used to investigate the correlation between radiation exposure and incidence of several non-infectious diseases in the local populations. It was hypothesized that searching dose–effect correlations in large population groups (from several thousands to several million inhabitants) could allow for identifying better the patterns of radiation/radionuclide exposure and health effects.

## 2. Materials and Methods

### 2.1. Sample Preparation and Analyses

The sampling of environmental materials and in situ measurements of radiological parameters was conducted in the period from 2011 to 2018 at Belokurikha town in Altai region, and at Kolyvan town in Novosibirsk region, both in Russia; in the Zhuhai city, Guangdong province, People’s Republic of China; and in the city of Echassières, Auvergne region, France (Figure 1).

Environmental materials sampled for analysis included rock and soil in order to determine the distribution of ^238^U and ^232^Th across the entire weathering profile of granites. Samples were collected from quarries throughout all layers of the upper crust profile, i.e., from the surface soil, through the weathering crust, and down to the granite bedrock. A 2 kg duplicate sample was collected from each horizon and dried for three days at room temperature. Then, each sample was put in a plastic container, carefully sealed, and transported within 1–3 days to the laboratory in Tomsk Polytechnic University (TPU). The sampling of granite rocks, weathering crusts, and soils and the sample preparation were carried out in accordance with the Russian State Standard 17.4.1.03-83 [41].

In the field, gamma-spectrometric analyses were carried out using a portable equipment SatisGeo GS-512 with scintillometer GSP-3, based on a NaI 3” × 3” detector, with a ^137^Cs test and calibration pad. The in situ gamma-spectrometric measurements were performed according to well-established guidelines for the identification of radionuclides [42]. The ambient gamma dose rate was measured at 1 m above the ground with the same instrument [6].

The activity concentrations of radon in surface air, or radon volume activity (RVA) measurements were made using the equipment “Alfarad” RRA-01M-01. The principle of device operation is based on the electrical deposition of ^218^Po (^222^Rn decay product) on the surface of a silicon detector and measurement of the ^218^Po alpha radiation. The radon flux density (RFD), i.e., the ^222^Rn transfer rate per unit area from the ground to the atmosphere, was determined as a measure of the Rn source strength. The determinations of RVA (Bq m^−3^) and RFD (mBq (m^2^ s)^−1^) were based on certified reference methods [43]. The radiometric methods and guidelines followed are compatible with ISO 11665-11:2016 [44]. The detection limit for RVA in the air was 1 Bq m^−3^. The maximum relative standard error of RVA determinations was 20%.

The mineralogical and geochemical analysis of granite, weathering crust, and soil samples were performed in accredited laboratories of TPU using certified methods, such as instrumental neutron activation analysis (INAA), fragmental radiography (f-radiography), X-ray diffraction analysis (XRD), and scanning electron microscopy (SEM).

The INAA was used to investigate the elementary composition of granite, weathering crust, and soil samples. The research was carried out at the IRT-T research reactor in the nuclear geochemical laboratory of TPU, irradiating samples in a permanent vertical channel with thermal neutrons with an integral dose 2 × 10^17^–1.5 × 10^18^ n/cm^2^ (certificate of accreditation No. ROSS RU.0001.518623 dated 10 October 2011). The INNA method does not require the chemical preparation of the sample, as it is based on the gamma spectrometric analysis of radioactive isotopes formed during the neutron bombardment of samples. For this analysis, the geological material was grinded to 100 mesh, and sample aliquots of 100 mg were wrapped in aluminum foil for neutron irradiation. As a method quality control, standard samples were irradiated simultaneously and under the same conditions with the samples under study. The contents of chemical elements were determined by comparing the radiation intensity of samples and reference samples in selected gamma photopeaks. This method has a high sensitivity and the detection limit was 0.06 g/t for U and 0.01 g/t for Th. The relative standard error in element determination was 5–15%.

The f-radiography method shows the distribution of U fission tracks in soil or clay samples. The basis for the f-radiography method is the nuclear fission of U atoms by thermal neutrons and registration of the fission fragments on a mica (phlogopite) detector. The detector registers the traces from fission fragments (tracks), which are counted under an electronic microscope. The number of fission fragment tracks is proportionate to the U content of geological samples and, through appropriate calibration curves, these are converted in concentrations (g/t). These analyses were performed at the IRT-T research reactor in TPU.

Further analyses were carried out by XRD and SEM on grain-size fractions of soils (grain size fractions of 1–0.5; 0.5–0.25; 0.25–0.01; and 0.01–0.04; <0.04 mm) separated by the analytical sieve set EAH2.1, compliant with DIN ISO 3310.

The XRD was used to determine the mineral composition of granites, weathering crusts, and soils. The method is based on the ability of flat grids formed by atoms in the crystal lattice of a mineral to reflect an incident X-rays beam, which leads to the appearance of diffraction reflection patterns. Bruker’s D2 Phaser diffractometer with the implementation of X-ray imaging in the Bragg–Brentano geometry was used for analysis. For XRD, the test sample was crushed to powder and placed in a quartz glass cuvette. According to the standard method applied, the following shooting parameters were used: Cu anode, X–ray tube voltage was 30 kV, current was 10 mA. The shooting angles 2Ɵ in the gross analysis of the sample composition ranged from 5° to 100°, rotation was 10 rpm, shutter speed was 1.5 s at a point, and the step was 0.02°. The Eva software package based on the PDF 2 X-ray powder diffractometric databases of the International Diffraction Data Center (ICDD, Denver, CO, USA) was used to decipher the radiographs.

The study of the mineral phases of chemical elements in samples was conducted using Hitachi S-3400N scanning electron microscope with energy dispersive spectrometer Bruker XFlash 4010.

The analytical work carried out to characterize the study regions included the analysis in laboratory of 37 solid samples (8 granite, 14 clay, and 15 soil samples) by several techniques, 53 in situ gamma-ray spectrometry analysis and radon measurements. Software packages Statistica 13.0 and Microsoft Excel were used for statistical data processing.

### 2.2. Calculation of Effective Dose

The annual effective doses received by the inhabitants of the four regions were calculated using the data from ambient radiation dose measurements, and activity concentrations of radon (^222^Rn) in outdoor and indoor atmosphere (Bq/m^3^) through the equation
Ef = CRn × DCF × FRn × OF × T(1)
where
CRn is the radon concentration in the air (Bq m^−3^);DCF is the dose conversion factor for inhaled ^222^Rn and equal to 9 nSv Bq^−1^ h^−1^ m^3^ [38];FRn is the attached fraction of Rn progeny and assumed as 0.8;OF is the occupation factor, and made equal to 0.4 for indoor and outdoor occupation;T is exposure time (8760 h y^−1^).

Due to the scarcity of data for thoron concentration in indoor and outdoor air for most areas and the lack of information on radionuclides in the population diet of these areas, the effective dose from thoron inhalation and radionuclide ingestion could not be estimated. Therefore, the effective doses calculated in this paper are an underestimation of the true value for total effective doses. Nevertheless, based on studies performed in other HBRAs, the contribution of radionuclides ingested with food and water is a very small percentage of the total effective dose, whereas the contribution of ^220^Rn inhaled may be similar to the contribution of ^222^Rn. Furthermore, the dose from external radiation is around 1/3 of the total, while ^222^Rn and ^220^Rn inhalation may contribute to about 2/3 of the total effective dose [37].

### 2.3. Public Health Data Analysis

Data on the morbidity rate for lung cancer, nasopharyngeal carcinoma, leukemia, and birth defects (years of 2014–2016) in the Belokurikha town, Altai region, was provided by the central district hospital of Belokurikha. The data on morbidity rate for the same diseases (years of 2011–2016) in Kolyvan, Novosibirsk region, was provided by Kolyvan central district hospital. For the other regions, the data were extracted from medical statistics presented in reports by Russian and foreign authors, and from official State reports on the status of sanitary and epidemiological welfare of the population [45,46,47,48,49,50,51,52,53,54,55,56,57,58,59,60,61,62]. The morbidity rates were adjusted for two age groups: children (0–14 years) and adults (aged 18 years and above).

### 2.4. Description of the Geology of the Study Regions

All studied regions are characterized by the occurrence of highly radioactive granites.

#### 2.4.1. Belokurikha, Altai Region, Russia

Coordinates: 51.985927 northern latitude, 84.916044 eastern longitude. The intrusive granite complex of Belokurikha constitutes a part of the western Altai-Sayan folded area at the orogenic belt in the structure of Central Asia. Belokurikha’s pluton is confined to the transition zone between the Biysk–Barnaul depression and the orogenic belt. The Iskrovsko–Belokurikha U and rare metals ore zone is confined to the tectonic zone with the same name, which features an east–west oriented fault system. This intrusive complex is characterized by elevated ambient gamma dose rate and Rn prone areas [63]. Endogenic U deposits were identified in this region, as well as deep aquifers with high concentrations of dissolved radon that are exploited to supply water for balneal purposes at the Belokurikha spa [64].

#### 2.4.2. Kolyvan, Novosibirsk Region, Russia

Coordinates: 55.354092 northern latitude, 82.767489 eastern longitude. Kolyvan is located within the Ob granitoid massif, in the south-eastern part of the West Siberian Plain, at the junction of the plate with the same name with the mountain structure of Altai-Sayan folded area. The Ob granite massif covers an area of 22 km^2^. The Skalinskoye granite formation is located 5 km north of the town of Kolyvan. This formation, with granite reserves accounting for 13,797 thousand m^3^, houses quarries of granitic stones mined for use as construction materials in the Novosibirsk region [65].

#### 2.4.3. Zhuhai, Guangdong Province, China

Coordinates: 22.321969 northern latitude, 113.543242 eastern longitude. The Guangdong Province is located within the west arc along the Pacific metallogenic ore belt and it is confined to the folded juncture of the Yangtze and South China plates. The rock foundation underneath Zhuhai city and its suburbs is composed of biotite, and porphyritic and monzonite granites. Due to its humid climate, which induces intensive weathering of rocks, the granites are covered by a well-developed weathering crust with a thickness up to 50 m. In the Guangdong Province, there are ion-adsorption type rare-earth element (REE) deposits formed in these granite weathering crusts [66,67].

#### 2.4.4. Echassières, Auvergne Region, France

Coordinates: 46.177037 northern latitude, 2.955052 eastern longitude. The Massif Central of France lies between the Rhone, the Garonne, and the Loire River basins. This mountain area was shaped from an ancient mountain range formed during the Hercynian Orogeny. The northern part of the Massif Central (Auvergne nucleus) is composed of crystalline schists folded at the end of the Precambrian and intruded by other granites, including the Beauvoir granites. The Echassières granite samples were collected within the Beauvoir granitic complex [68].

## 3. Results

### 3.1. Geochemical Characteristics

The concentrations of primordial NRE in samples, determined by INAA, are given in Table 1. The results show that the concentrations of the natural radioactive elements U and Th in these four regions are consistently much higher than the global average concentrations in the Earth’s crust, and thus confirmed the exceptional characteristics of the selected regions as high natural radiation background regions. It must be noted that the natural radioactivity in the areas of Zhuhai city (Guangdong Province), Belokurikha town (Altai region), and Kolyvan town (Novosibirsk region) primarily arises from thorium and thorium decay products (Th/U >> 1), while the natural radioactivity in the area of the Échassières town in the Auvergne region is mostly from uranium and uranium decay products (Th/U = 0.1).

The analysis of chemical and mineralogical composition and other characteristics of rocks were made to investigate the formation process of weathering crusts and autochthonous soils, and the re-distribution of natural radionuclides from the crystalline bedrock into other layers (Figure 2). The chemical composition of autochthonous soils was largely inherited from the chemical elements present in parent rocks. However, depending on the weathering conditions of each region (e.g., climate, Eh, pH, and presence of clay minerals), the NRE were re-distributed in the weathering profiles in a different manner. Nevertheless, the maximum concentrations of U and Th were observed always in clay horizons (Table 1).

The speciation of primordial NRE, especially U, changed during the process of granite weathering. In general, with weathering the crystalline rockbound NRE become more mobile and ended up adsorbed onto clay minerals (kaolinite, montmorillonite, etc.). From the results of analysis made on the soil grain-size fractions and fragmentation radiography, it was observed that the main concentration of U occurred in the finer grain fractions (0.01–0.04 mm; <0.04 mm) of weathering crusts and soils. This resulted from the re-concentration of uranium released from granites through sorption of uranium ions onto clay minerals [71]. Therefore, over geological times, U and Th and their progenies were transferred from the crystalline rock (where their concentrations were more moderate) to granitic subsoil and clay zones in the weathering crust profiles and became more concentrated in the upper layers enriched with kaolinite and montmorillonite. This was investigated in detail in the Belokurikha and Kolyvan soil-rock profiles (Figure 2). A similar process was reported for granites and soils in the Guangdong Province [72].

Through this process, clay in soils became the main deposit of primordial radionuclides and the source of their radioactive progeny in soil surface horizons. However, clay is not an active sorbent for Rn and this radioactive noble gas, once formed from radioactive decay of Ra, is able to migrate by diffusion in soil pore fluids up to surficial soil horizons and generate enhanced Rn concentrations near the soil surface (Figure 2). The same was observed with thoron gas (^220^Rn), from the thorium radioactive decay series, which was present in high concentrations in thorium-rich soils and particularly in clay layers in the Guangdong Province [72].

The results of a comprehensive radiological survey carried out at Belokurikha town and adjacent settlements (expedition “Berezovgeologia”) showed that RVA (^222^Rn) values in soil gas spread over a wide range, of 1–120 kBq m^−3^, and in Kolyvan of 630–1570 kBq m^−3^ [73,74].

The transfer of U and Th from crystalline granite (where radon exhalation was reduced) to clay and soil layers did enhance the exhalation of radon isotopes into the soil pore spaces and increased the radon flux into the atmosphere. Therefore, exposure to radon isotopes was expected to be higher in regions with soils formed from weathered granites in comparison with exhalation from crystalline rocks. More detailed reports of the chemical and mineralogical composition and distribution of NRE in profile horizons are given elsewhere [1,71].

Radon exhalation rates from monazite sands has been reported to be much lower than in soils and this explains the relatively lower concentrations of radon (^222^Rn and ^220^Rn) in surface air at Kerala, India, despite the high concentrations of Th and U in the mineral sands. Insights on the radon exhalation from minerals were provided by Krupp et al. [75].

### 3.2. Radiological Characteristics

The high Rn concentrations, high ambient radiation dose rates, and other radioecological parameters determined in the study areas stem from geological formations composed of granites with high concentrations of primordial radionuclides, as seen above. The radionuclides in these rocks, weathering crusts, and soils are at the origin of high ambient gamma ray dose rates, intensive radon exhalation from the ground, and high ^222^Rn concentrations in surface air in uranium-rich areas. In thorium-rich areas, there are also high ^220^Rn (thoron) concentrations in surface air. In both uranium- and thorium-rich areas, human exposure may be further enhanced by the use of local geological materials with high NRE concentrations in building construction [6,7,76]. Furthermore, in these regions, natural waters with a high content of Rn, Ra, U are used for human consumption and may contribute to enhance the radiation exposure of populations through ingestion [77].

The data on Rn obtained with the radon analyzer “Alfarad” in September 2017 in the areas of Belokurikha and Kolyvan, together with the results of investigations carried out in the area of Zhuhai city [72], the data concerning the area of the town of Echassières [78], and the data on ^222^Rn flux density are presented in Figure 3. In general, within the study areas, the average levels of RVA (^222^Rn) exceeded the acceptable radon level approved in the Russian Federation, of 100–200 Bq m^−3^. For comparison, the International Commission on Radiation Protection recommended a reference level of Rn concentration in dwellings in the range of 100–300 Bq m^−3^ [39]. The European Union recently adopted 300 Bq m^−3^ of Rn as the recommended maximum Rn level in indoor air [34], but the ultimate aim is to decrease this value to 100 Bq m^−3^, as advised by WHO [33].

The buildings in the Belokurikha town and adjacent settlements displayed high RVA levels indoors, 284–560 Bq m^−3^ [79]. RVA levels in the indoor air of Kolyvan town were up to 400 Bq m^−3^ [73,74].

**Figure 3 ijerph-19-08643-f003:**
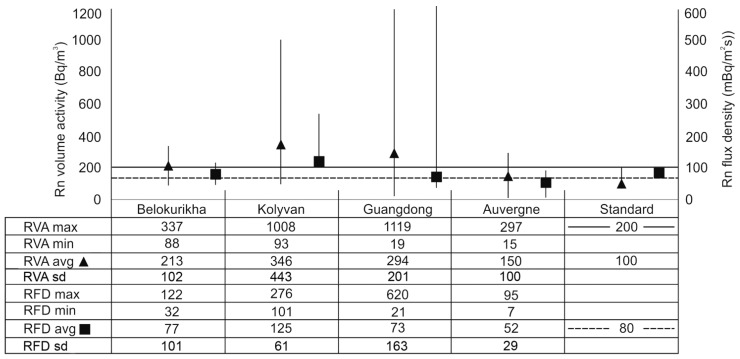
Graphic display and numerical data from measurements of RVA (Bq m^−3^) and RFD (mBq (m^2^ s)^−1^) in the study regions. RVA standard according to Radiation Safety Standards (NRB-99) [33,35]; RFD standard according to Practical Guidance 11-102-97 [80]. Avg, average; sd, standard deviation.

In the case of RFD, or radon exhalation rate from the soil surface, the reference value set by Russian standards, 80 mBq (m^2^ s)^−1^, was exceeded in the town of Kolyvan in Novosibirsk Oblast (Figure 3). On itself, this would recommend using Rn-mitigation measures to prevent the Rn inflow into buildings [80]. In all other study regions, the average values of RFD were lower than 80 mBq (m^2^ s)^−1^. However, within large provinces or towns with average RFD values lower than the adopted standard, the values are not uniform and many places display extremely high values, such as, for example, 620 ± 76 mBq (m^2^ s)^−1^ in the area of Zhuhai city, China. In the Guangdong Province, the Rn concentrations in soil gas ranged from 18 to 960 kBq m^−3^ [81]. In such cases, according to the Russian standard SP 11-102-97, Rn countermeasures should be applied in the construction and be a part of the civil engineering project from the beginning. Notwithstanding, it was found out that in the southern regions of China, especially in the Guangdong Province, the buildings are constructed with bricks made from local clay, which has a high radioactive content (U = 8.5 ± 0.3 g/t, Th = 53.4 ± 4.6 g/t), and further increases the radiation exposure of local population in their homes.

A high activity of thoron (^220^Rn), a daughter product of ^232^Th, can be an additional factor of significant radiation exposure for the local population in Zhuhai, China. Because of its short half-life (T_1/2_ = 55 s), ^220^Rn has for long time not been considered a radiological hazard, although some recent reports give increased attention to ^220^Rn as a radiation hazard for humans [37,76,82,83]. In the South China provinces, due to the distribution of thorium-containing rocks and soils, as well as the use of clay and crushed stone rich in ^232^Th as construction materials, there are high activity concentrations of ^220^Rn in indoor air, which are non-negligible from the standpoint of radiation dose [72]. Unfortunately, for most of the HBRAs, there are no data on ^220^Rn concentrations.

The contributions of various pathways for the total effective dose are displayed in Table 2. There are limitations in the information available for each area, but the data are robust in showing that radon through inhalation is the most important contributor to the dose. When data are available both for ^222^Rn and ^220^Rn (thoron), thoron contribution can be as important as ^222^Rn contribution to the effective dose. This means that dose assessment for areas made through ^222^Rn measurements only have been underestimated [37]. The contribution of external radiation to the effective dose is much lower than the contribution of radon inhalation (Table 2). The contribution of radionuclides ingested with diet and water was not evaluated yet, but based on other studies, probably it will be minor. This pathway analysis and their relative importance suggest that previous studies in HRBAs, especially those based on measurements of external radiation only, might have underestimated the effective dose received by members of the population.

### 3.3. Effective Doses

Table 2 shows the estimated effective doses for the inhabitants of the four HBRAs based on radon inhalation indoor, radon inhalation outdoors, and external irradiation. The average values for these HBRAs range from 5.6 ± 1.7 to 23.0 ± 8.6 mSv y^−1^, while for the most exposed members of the population in the same areas, the effective doses range from 9.7 ± 1.7 to 57.8 ± 8.6 mSv y^−1^. These are high radiation doses, in the range of values reported for worldwide HBRAs.

### 3.4. Analysis of Statistical Data on Disease Incidence Rates

To assess the effects of high radiation background on the health of local populations, we used as biological endpoints the morbidity data on malignant neoplasms (MN) of lung, nasopharynx, hematopoietic tissue (leukemia), and congenital malformations of the fetus (CMF) or birth defects. To calculate the incidence rate, the number of first-time reported cases per year was divided by the average annual population of the study region (city) and multiplied by 100,000 people to express incidence rate as (^0^/_0000_). It was previously shown that radiation exposure has a greater impact on children’s health compared with adult health [6,7,8,9,10,11] and, to assessing this impact, Table 3 shows cancer incidence rates discriminated by age groups.

#### 3.4.1. Belokurikha, Altai Region, Russia

The population of the Belokurikha town from 2014 to 2017 was 14.5–15.3 thousand inhabitants. Statistical records for this region showed that the incidence rate of childhood cancer is 2.8 times higher in the Belokurikha town compared with Russian global statistics (Table 3). For example, in 2015, the infant incidence of leukemia in Belokurikha, with a value of 7^0^/_0000_, was two-fold the national average. Furthermore, in 2015, the incidence of cerebral malignant neoplasms in children aged from 0 to 14 was 35^0^/_0000_, while the all-Russia average was 2.1^0^/_0000_ [62]. Based on these indicators, the Belokurikha town was classified by the sanitary authorities as a high-risk area with respect to incidence rates of respiratory diseases (bronchitis, asthma, etc.), diseases of the hematopoietic system, type 1 diabetes mellitus, and primary childhood disablement [45].

Often, it is the deterioration of the quality of environment caused by industrial activities (for example, air pollution and contamination of drinking water) that poses hazards to public health and causes increasing number of occurrences of non-contagious diseases. However, it must be underlined that in Belokurikha the air pollution levels meet the air quality and hygiene standards, and there has been even a trend in air quality improvement over the years (the minimum air pollution levels were recorded in 2014–2016). The quality of drinking water in this town also met all sanitary and epidemiological standards. Moreover, no toxic chemicals other than NRE have been identified in the area [45].

Therefore, for the population of the Belokurikha town, it seems plausible that it is the high RVA outdoors (up to 337 ± 102 Bq m^−3^ outdoors, see Figure 3) and the even higher RVA indoors (284–560 Bq m^−3^), combined with the elevated levels of ambient radiation up to 350 nGy h^−1^ (35 mR h^−1^) that pose a serious hazard to developing respiratory diseases and cancer. For comparison, the world average outdoor absorbed dose rate from environmental gamma radiation is 59 nGy h^−1^ and the worldwide range is 18–93 nGy h^−1^ [6,11].

#### 3.4.2. Kolyvan, Novosibirsk Region, Russia

The population of the Kolyvan town from 2011 to 2016 was 11.8–12.5 thousand inhabitants. In the Kolyvan town, the general cancer incidence rates among children and adults are, respectively, 4 and 1.5 times higher than the Russian national average. In addition, there is a high incidence rate of lung cancer (9 times higher than the Russian national average), high leukemia incidence (7 times higher than the Russian national average), and congenital defects (1.5 times higher than the Russian national average). In 2015, a high index of testicular cancer reaching 26^0^/_0000_ was observed among the children aged 0 to 14, while in Russia, the average incidence does not exceed 1^0^/_0000_ [62]. In general, the Kolyvan district has also an elevated incidence of non-infectious diseases when compared to other areas of the Novosibirsk region [47].

Searching for the potential causes of these high incidence rates of cancer and other diseases in the Kolyvan population, high radioactivity in the atmosphere and in the local environment appears first as a candidate. Indeed, exposure to radioactivity is high due to enhanced levels of radon originated from open quarries of granite located in close proximity to the town (4 km). These granites contain 9.6 ± 0.8 g/t of U and 34.0 ± 2.9 g/t of Th (Table 1), which generate elevated levels of RVA in surface air, up to 1008 ± 443 Bq/m^3^ outdoors (Figure 3) and RVA in indoor air, up to 400 Bq/m^3^. In addition, the use of local granite as ornamental rock and crushed granite stone as a bulk construction material in residential buildings explains the current level of ambient radiation in town, up to 500 nGy/h (50 mR/h), much above the worldwide average ambient radiation dose rate of 59 nGy/h [6,11].

#### 3.4.3. Zhuhai, Guangdong Province, China

The population of the Guangdong Province from 2008 to 2018 was 104.41–123.48 million and of Zhuhai city 1.29–1.67 million inhabitants. According to WHO, the South-Eastern Asian provinces are the areas with the highest nasopharyngeal malignant neoplasia (MN) incidence in the world [84], with 11–25 cases of nasopharyngeal MN detected in the Guangdong Province for both sexes per 100,000 inhabitants [48,58]. In Russia, this indicator ranges from 2 to 7^0^/_0000_ [54] at the national level, and in the world, it ranges from 2 to 4^0^/_0000_ [49,57]. The incidence of congenital malformations of the fetus (CMF), MN incidence among children, and lung cancer incidence among adults also ranked high in the Guangdong Province (Table 3).

In some areas of this province, extremely high values of RVA and RFD were measured (Figure 3) and ^222^Rn concentrations in groundwater used for human consumption reached up to 1980 Bq L^−1^ [77]. The construction of local houses makes a heavy and widespread use of bricks made from local radioactive clay, which contains 8.5 ± 0.3 g/t of U and 53.4 ± 4.6 g/t of Th (Table 1). Thoron gas (^220^Rn), a decay product of ^232^Th, is an additional radiation hazard, reported as non-negligible from the standpoint of radiation dose [72].

#### 3.4.4. Echassières, Auvergne Region, France

The population of the Auvergne region from 2007 to 2018 was 7.41–7.99 million inhabitants. In the Auvergne region, the incidence rate of all types of cancer is three times higher than the worldwide average in all age groups [49]. Lung cancer incidence is also higher compared with average incidence rates worldwide and France [50]. The high content of U in Echaussières granites (18 g/t) and, as a result, the high RVA in the air (Figure 3), pose radiation risks that are considered to affect the morbidity rate of people in the Auvergne region (Table 3).

From the above, it seems clear that incidence rates of cancer and several non-infectious diseases are noticeably increased in each study region when compared with national and worldwide average morbidity data. No potential causes for increased morbidity, other than the natural high radiation background of the regions, were identified. Furthermore, plotting the morbidity rates of the four study areas against the radiation exposure, showed a clear proportionality between dose and effects for most exposure pathways, as discussed in the next section.

Cancer and other health effects can be induced by chronic exposure to high radon concentrations (through inhalation) and chronic exposure to ambient radiation (external irradiation). Furthermore, the environmental transfer of naturally occurring radioactive elements, such as ^226^Ra, ^228^Ra, ^210^Pb, and ^210^Po, may also contribute through ingestion of food and water to the total radiation dose received by humans. In many regions, it was observed that NRE released from the crystalline rock end up to entering trophic chains following pathways, such as *rock → water → soil → plants → animals → humans*. Altogether, through several transfer pathways, the NRE may reach human beings and originate a substantial increase in radiation exposure with potential effects on human health [36,85,86,87].

The assessment of radiation exposure through ingestion was not in the scope of this study, but it may further enhance the radiation doses received by humans through inhalation and external irradiation in the selected study areas and might explain some of the variation in public health data among the four regions investigated. Critical radionuclides, i.e., those that contribute more to the effective dose, and the relative importance of transfer pathways, may vary also among regions. It is worth mentioning that a recent investigation in a high natural radiation zone in Bat Xat, Vietnam, related to rare-earth surface deposits reported total annual effective doses to members of the most exposed population group averaging 38 mSv y^−1^ [37]. A detailed study on the radiation exposure pathways in that area showed that, to such total effective dose, the radionuclide inhalation (^222^Rn and ^220^Rn isotopes) contributed with 70%, the ambient gamma radiation from radionuclides in the ground contributed with 29%, the cosmic radiation contributed with 0.5%, and the ingestion of local foods and water contributed with 0.5% [37]. Another study for an HBRA in Mamuju, Indonesia, reported a similar percent contribution of exposure pathways and radon isotopes to the average effective dose for inhabitants, determined at 32 mSv y^−1^ (range 17–155 mSv y^−1^) [88]. Detailed studies such as these ones are missing for most of the HBRAs referred above. The exposure pathways may vary among the four HBRAs investigated, but the recent studies in Vietnam and Indonesia highlight the overwhelming importance of radon (both ^222^Rn and ^220^Rn) to the dose in HBRAs.

### 3.5. Analysis of the Dependence of Morbidity Rates on Radioecological Parameters

Morbidity rates in all selected study regions were elevated, although there were differences among the four regions as well as there were differences among radiation levels and concentrations of the radionuclides in the environment.

In brief, Belokurikha and Kolyvan were areas with enhanced incidence rates of all types of cancer for children aged 0 to 14; Kolyvan was also an area with enhanced incidence rates of lung cancer and leukemia; the Guangdong Province was a high-incidence area for nasopharyngeal carcinoma. There was a high incidence of birth defects in Kolyvan and the Guangdong Province.

The dependence of morbidity rates of different types of cancers and CMF on radioecological parameters in the study regions is shown in Figure 4, where correlations are established, although not always with a high significance level (*p* < 0.05). A positive correlation was established between the increase in incidence rate of all types of child cancer and RVA (Figure 4A). Interestingly, the dependence of child cancer morbidity on the sum of U and Th concentrations in the granite showed no correlation, which means that morbidity is not simply related to the concentration of these elements in the Earth’s crust. This highlights that a significant exposure to radiation (in this case, from radon) requires the transfer of radioelements from the bedrock to the biosphere, and radionuclide transfer pathways may or may not exist locally (Figure 4B). The increase in the level of birth defects or CMF is again positively correlated with the RVA (Figure 4C), and it was similar to the trend in concentrations of U and Th in granites (Figure 4D). There was a tendency to increasing incidence of lung cancer in the whole population with a rising RVA, in contrast to U concentration in granites, which showed again that exposure (radionuclide transfer pathways to humans) is more important than NRE concentrations in the ground (Figure 4E,F). Leukemia incidence rate in children and in all age groups of the population seemed to depend strongly on RVA levels (Figure 4G,H). The increase in the incidence of nasopharyngeal carcinoma was strongly correlated with Th concentrations in soils, therefore through ^220^Rn (thoron) inhalation (Figure 4J), but weakly correlated with the RVA for ^222^Rn (from ^238^U) in the air (Figure 4I). It should be noted that lymphoid tissue is a major site for the accumulation of Th in the human body (e.g., from inhaled dust), and a large amount of this tissue is especially concentrated in the nasopharynx. Moreover, the radiological impact of the inhaled short-lived thoron (^220^Rn from ^232^Th) is likely much more significant in the nasopharynx (upper part of the respiratory tree) in comparison with the longer-lived ^222^Rn and its progeny that affects mainly the deeper portions of the respiratory tree.

Globally, the graphic plots in Figure 4 show that morbidity is hardly correlated directly with U and Th concentrations in the geological deposits, and this is probably a main reason why many studies failed in the attempt to demonstrate the association between radioactivity levels in geological deposits and biological effects. Several graphic plots in Figure 4 clearly show that, when the transfer of radioactivity to humans (exposure pathway) is considered, in this case the ^222^Rn and/or ^220^Rn inhalation, there is a statistically significant positive correlation between radiation exposure and the observed health effects.

It is recognized that it is difficult to establish in an unambiguous manner the contribution of natural radioactivity to the overall set of factors affecting morbidity rates. However, in regions with a high radiation background, such as those investigated in this paper, there are associations between the incidence of cancer and other non-infectious diseases with the high environmental radioactivity levels, and specifically with radon. It is worth noting that in some of these regions no other potential hazardous factors (atmospheric, soil and water pollution) were identified. The four study areas have different radioactivity and radiation levels, but when the data from the study areas are plotted together, the trend of increasing morbidity rates with increasing radiation exposure is striking. This is further highlighted through the comparison of the morbidity rates of the study regions with the national or worldwide health statistics. Therefore, natural radiation levels in these study regions are pointed out as the probable cause of high incidence rates of cancer and other non-infectious diseases observed in their respective populations.

Radon (either ^222^Rn or ^220^Rn, or both) was present always in these HBRAs, in U- and Th-rich areas, and the classification in «radon-prone» and «non-radon» HBRAs as proposed by some authors is not justified [4]. Indeed, the results indicate that radon isotopes (^222^Rn and ^220^Rn) are the critical radionuclides both in U- and Th-rich areas (Table 3).

## 4. Data Uncertainties and Limitations

The data used in this study display several limitations and uncertainties. This paper includes measurements of radon activity concentrations in outdoor air and ambient dose rates and analysis of radioactive elements in rocks and soils. However, information about ^222^Rn and ^220^Rn activity concentrations in indoor air is not available for all research regions, although it is known that a large part of radiation exposure may come from radon. In addition, drinking water radioactivity, which might be also an important factor in health impact, is considered very briefly. We are working on obtaining these data.

Medical statistics on population morbidity are presented in a much-generalized form. It is needed a more detailed division of population cohorts by gender, age, lifestyle, type of occupation, etc. When comparing the results with the worldwide and national averages, statistical significance tests were not applied. The small amount of data on morbidity in the Auvergne region is due to the limited information available. Finally, this study does not account for other potential hazardous sources of human exposure (smoking, alcohol consumption, Epstein–Barr virus, air, soil and water pollution, etc.) that can cause several types of cancer.

These preliminary results are expected to stimulate in-depth research, including comprehensive radiation monitoring, especially the measurement of indoor Rn (^222^Rn and ^220^Rn) concentrations, along with the assessment of radiation exposure through the local diet. Furthermore, because local materials (clay, granite, crushed stone, etc.) are used in building construction, it is necessary to regulate the use of such materials and introduce the spectrometric analysis of materials to license their use. The construction of buildings in areas with high background radiation may require also systematic implementation of Rn preventive and Rn countermeasures to reduce radon exposure and curb cancer incidence rates.

## 5. Conclusions

The results of this empirical study embracing four HBRAs in granitic regions show that the geochemical and weathering processes intervening in soil formation made the transfer and re-concentration of primordial radionuclides from crystalline rock into the upper soil layers enriched also with kaolinite and montmorillonite. The emanation of radon isotopes from the ground into surface air was, therefore, facilitated by this process and originated the high radon concentrations in outdoor and indoor air in such regions.

The populations of these areas are exposed to much higher radiation doses in comparison with the worldwide average exposure, both for external irradiation and internal irradiation. The main exposure pathways were radon (and probably thoron) inhalation, followed by external irradiation and, in a minor degree, by the ingestion of radionuclides with water and foods. The effective doses for the most exposed individuals varied from 9.7 ± 1.7 mSv y^−1^ in Echassières to 57.8 ± 8.6 mSv y^−1^ in the Kolivan town, estimated based on external dose and radon inhalation (^222^Rn). Total exposures in these areas are considered high, but when data for thoron (^220^Rn) becomes available, probably the estimated total effective doses will further increase by about 1/3.

Analyzing public health statistics for the HBRAs and comparing them with national and global health statistics showed that the morbidity in HBRAs was clearly higher in the HBRAs for non-transmissible diseases and cancers. These comparisons assume that limitations or bias in obituary and statistical records may exist, but would be similar within the same country. Furthermore, the four HBRAs have different radiation levels and the correlations of radiation exposures with morbidity rates indicated that the dose–effect relationships are positive and linear within a reasonable statistically significance level (*p* < 0.05).

Radon appears as the critical radionuclide, contributing through inhalation to the effective dose much more than the external exposure and correlating significantly to cancer, leukemia, and borne defects in the population. Furthermore, thoron seems highly correlated with nasopharyngeal cancer morbidity.

The radiation exposure levels in Belokurikha and Kolyvan (Russia), Zhuhai (China), and Echassières (France) enlarge the range of dose rate values previously reported for Kerala, the large part of Ramsar (Iran), Xinjiang (China), and Guarapaguari (Brazil). Such dose rate values encompass high, as well as low, radiation levels in domains where the confirmation of health effects in humans has been controversial and the LNT model was questioned.

Based on these conclusions, it seems justified and necessary to implement renewed research in HBRAs with a better design, including a more complete characterization of radiation exposure and with sufficient statistical power to clarify the occurrence of biological effects in human beings. The results of such studies will be of high value for:Finally, clarifying whether there is a need to implement radiation protection measures in HBRAs to protect the public health;Learning from epidemiological studies in HBRAs in order to extrapolate lessons into the planning of remediation measures of areas affected by human activities, such as uranium mining areas and areas contaminated by nuclear accidents;Learning more about the effects of low doses in order to apply knowledge in the radiological risk assessment of medical exposures and of prolonged radiation exposures during space missions.

## Figures and Tables

**Figure 1 ijerph-19-08643-f001:**
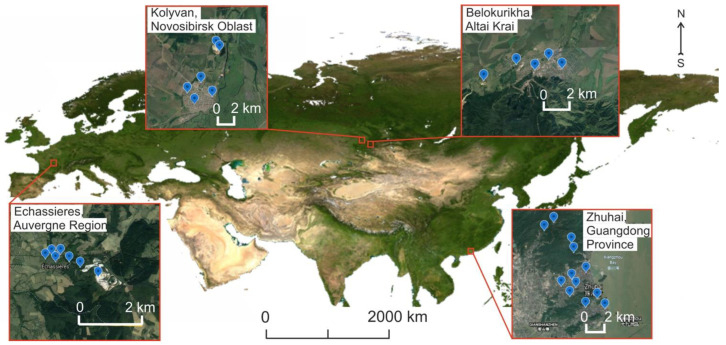
Study areas identified with city names and sampling sites (symbols)

**Figure 2 ijerph-19-08643-f002:**
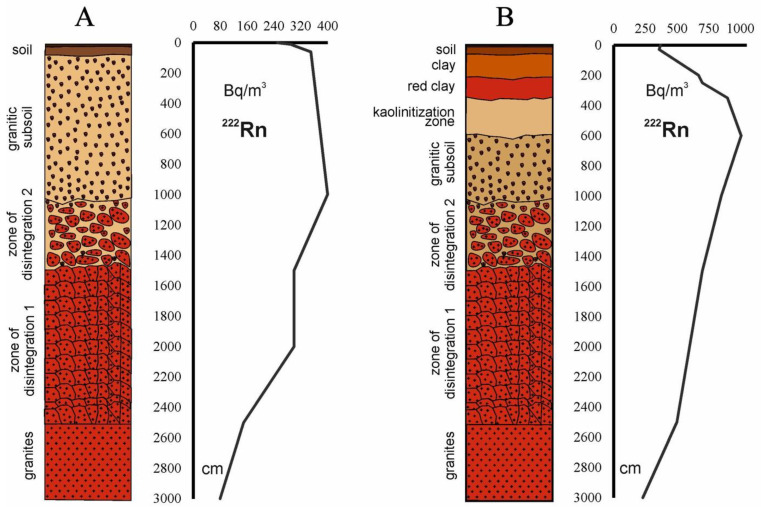
Distribution of ^222^Rn activity concentration in the interstitial volume with depth in the layers of granite, weathering crust, clay, and soil in the regions: (**A**)—Belokurikha, Altai region; (**B**)—Kolyvan, Novosibirsk region.

**Figure 4 ijerph-19-08643-f004:**
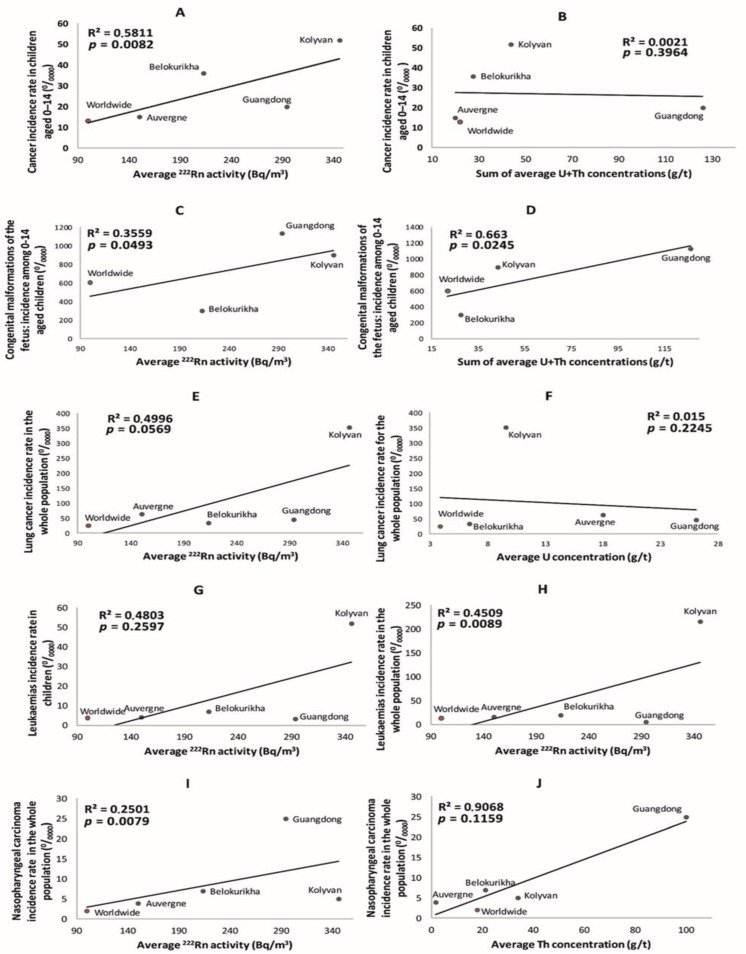
Dependence of all types of cancer incidence rate for children (^0^/_0000_) on ^222^Rn activity (Bq m^−3^) (**A**), and on the sum of U and Th concentrations in granites (g/t) (**B**). Dependence of CMF incidence rate for children (^0^/_0000_) on ^222^Rn activity (Bq m^−3^) (**C**), and on the sum of U and Th concentrations in granites (g/t) (**D**). Dependence of lung cancer incidence rate for the whole population (^0^/_0000_) on ^222^Rn activity (Bq m^−3^) (**E**), and on U concentration in granites (g/t) (**F**). Dependence of leukemia incidence rate for children (^0^/_0000_) (**G**) and for the whole population (^0^/_0000_) (**H**) on the ^222^Rn activity (Bq m^−3^). Dependence of nasopharyngeal carcinoma incidence rate for the whole population (^0^/_0000_) on the ^222^Rn activity (Bq m^−3^) (**I**), and on the average concentration of Th in granites (g/t) (**J**). Reference level of ^222^Rn activity according to WHO [33]; worldwide averages of U and Th concentrations according to Grigoriev [69]; worldwide averages of cancer incidence rate for children according to malignant neoplasms [54]; CMF incidence according to Hoffman et al. [55]; worldwide averages of lung cancer incidence rate according to Jemal et al. [49]; leukemia incidence rate for children according to Aksel et al. [62], and for whole population according to Jemal et al. [49]; worldwide average of nasopharyngeal carcinoma incidence rate according to Tang et al. [57].

**Table 1 ijerph-19-08643-t001:** Concentration of natural radioactive elements (g/t) and thorium–uranium ratio in granites.

		Crystalline Granite Bedrock	Zone of Granite Disintegration	Granitic Sub-Soil	Clay	Soil
Belokurikha, Altai	U	6.4 ± 0.5	9.7 ± 0.9	8.0 ± 0.8	11.4 ± 1.8	8.6 ± 0.9
	Th	21.1 ± 3.8	36.5 ± 3.5	52.6 ± 4.4	58.8 ± 4.8	35.9 ± 3.8
	Th/U	3.3	4.4	6.5	5.1	4.2
Kolyvan, Novosibirsk	U	9.6 ± 0.8	15.6 ± 1.3	10.2 ± 1.0	10.6 ± 1.1	4.9 ± 0.3
	Th	34.0 ± 2.9	73.0 ± 5.9	47.1 ± 3.3	57.2 ± 4.6	14.9 ± 1.3
	Th/U	3.5	4.7	4.6	5.4	3.1
Zhuhai, Guangdong	U	26.1 ± 3.1	12.4 ± 0.9	8.0 ± 0.3	8.5 ± 0.3	8.1 ± 0.7
	Th	100 ± 9.5	50.4 ± 3.1	51.2 ± 4.6	53.4 ± 4.6	47.6 ± 2.7
	Th/U	3.8	4.1	6.4	6.2	5.9
Echassières, Auvergne	U	18 [68]	n.d.	n.d.	n.d.	6.5 ± 0.5
	Th	1.7 [68]	n.d.	n.d.	n.d.	4.6 ± 0.3
	Th/U	0.1	n.d.	n.d.	n.d.	0.7
Worldwide averages [69,70]	U	3.9	n.d.	n.d.	4.3	1
	Th	18	n.d.	n.d.	14	5
	Th/U	4.6	n.d.	n.d.	3.2	5

n.d.—no data.

**Table 2 ijerph-19-08643-t002:** Estimated effective doses for inhabitants of the four high natural radiation background.

	Belokurikha, Altai	Kolyvan, Novosibirsk	Zhuhai, Guangdong	Echassières, Auvergne
**External dose, mSv y^−1^**				
Average	3.1	4.4	1.6	1.8
Min	0.8	1.3	0.5	0.7
Max	3.5	5.3	2.6	2.2
Standard deviation	1.1	1.6	0.6	0.6
**Outdoor ^222^Rn Inhalation, mSv y^−1^**				
Average	5.3	8.6	7.3	3.8
Min	2.2	2.3	0.5	0.4
Max	8.4	25.2	28.1	7.5
Standard deviation	1.9	3.2	2.7	1.1
**Indoor ^222^Rn Inhalation, mSv y^−1^**				
Average	7.1	10.0	3.5	n.d.
Min	2.0	1.0	0.5	n.d.
Max	12.3	27.3	24.0	n.d.
Standard deviation	2.5	3.8	1.1	n.d.
**Effective dose, mSv y^−1^ (^222^Rn inhalation + External irradiation)**				
Average	15.5	23.0	12.4	5.6
Min	5.0	4.6	1.5	1.1
Max	24.2	57.8	54.7	9.7
Standard deviation	5.5	8.6	4.4	1.7

n.d., no data.

**Table 3 ijerph-19-08643-t003:** Summary of disease incidence rates per 100,000 inhabitants in the high radiation background regions investigated and data from wider regions for comparison. For each region, the years of statistical information used are indicated in parenthesis.

Disease Incidence Rate per 100,000 Inhabitants (^0^/_0000_)	Belokurikha, Altai Region, Russia (2014–2017)	Kolyvan, Novosibirsk Region, Russia (2011–2016)	Guangdong Province, China (2008–2018)	Auvergne Region, France (2007–2015)	Russia Average Value (2007–2018)	Worldwide Standard 2007–2018
Cancer incidence rate for all population	**451**	**420**	250 [48]	615 [50]	408 [54,56]	242 [49]
Cancer incidence rate for children aged 0–14	**36**	**52**	20 [48]	15 [51]	13 [54]	13 [54]
Cancer incidence rate for adults aged 18 and over	590	**3334**	591 [48]	n.d.	2400 [54,56]	n.d.
Lung cancer incidence rate for both sexes	33	**352**	45 [48]	**63** [50]	41 [54]	24 [49]
Nasopharyngeal carcinoma incidence rate for both sexes	7	5	**11–25** [48,58]	4 [50]	5 [54]	1–2 [49,57]
Leukemia incidence rate for all age groups combined	20	**216**	6 [60]	17 [52]	19 [54]	14 [49]
Leukemia incidence rate for children aged 0–14	**7**	**52**	3.2 [61]	4.2 [51]	3.7 [62]	3.8 [62]
Congenital malformations of the fetus (CMF)	300 [46]	**897**	**374–1129** [59]	n.d.	≈632 [53]	≈600 [55]

n.d.—no data. The bold lettering and gray-colored boxes represent increased incidence rates relative to Russia and worldwide average values. Source of data in square brackets [ ].

## Data Availability

The data presented in this study are available in references [45,46,47].

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
