# Peer review of "Impact of Environmental Radiation on the Incidence of Cancer and Birth Defects in Regions with High Natural Radioactivity"

_ijerph, 2022, doi:10.3390/ijerph19148643_

Round 1

Reviewer 1 Report

The article is well structured and is a valuable contribution to the research related to the epidemiological assessment in case of areas with an increased level of natural radioactivity. Much of the fact that radioactivity is treated quite differently, because from the legal point of view, only the dose above the natural background counts, which for some regions may be even two orders of magnitude higher than the global average.

For the evaluation of residents' doses, given that they spend most of their time indoors, the measurements taken in buildings, especially radon and thoron, are of the greatest importance. Measurements made as part of the project concerned the external environment. The results of these measurements can therefore be treated primarily as an indicator that these are indeed areas with an increased natural content of natural radioactive isotopes.

The first cause of cancer is smoking. Usually it doesn't matter to children. Have you included this factor into consideration for the epidemiological analysis in other cases, or have you basically compared your result to country average ?

87-88 – At this point it could be specified for what time, equilibrium and conversion factor this dose was assessed

200 – Please check again the formula. The dose conversion factor of 9 nSv*m3/(Bq*h) corresponds to equilibrium equivalent concentration. Therefore it should be multiplied by the equilibrium coefficient (not attached fraction), and measured radon activity concentration, and annual occupancy, and year expressed in hours. I did not found that conversion factor in IAEA 2014, but it is in United Nations Scientific Committee on the Effects of Atomic Radiation. UNSCEAR 1993. Report to the General Assembly, with Scientific Annexes. Why have you accepted the occupancy factor as 0.5 ? The recommended one is equal to 0.8 (UNSCEAR, 1993, IAEA, 2014), and the recommended equilibrium factor 0.4.

Author Response

RESPONSE TO THE REVIEWER â„–1

We would like to thank the Reviewer for the detailed and critical analysis of our article. We agree with comments and recommendations made, which have contributed to improve the manuscript. We to each comment below:

  1. The first cause of cancer is smoking. Usually it doesn't matter to children. Have you included this factor into consideration for the epidemiological analysis in other cases, or have you basically compared your result to country average?

We agree that smoking is the most important factor causing lung cancer. However, at this stage of our study, we were unable to obtain data on smoking habits per ager groups. We have basically compared our HBRA data with country and world average morbidity statistics.

  1. 87-88 – At this point it could be specified for what time, equilibrium and conversion factor this dose was assessed

We are not sure about the parameters used by WHO in their calculations. Therefore, we prefer to simply recommend/send the interested reader to the original bibliographic reference. Sorry, but we prefer to keep the text it as it is.

  1. 200 – Please check again the formula. The dose conversion factor of 9 nSv*m3/(Bq*h) corresponds to equilibrium equivalent concentration. Therefore it should be multiplied by the equilibrium coefficient (not attached fraction), and measured radon activity concentration, and annual occupancy, and year expressed in hours. I did not found that conversion factor in IAEA 2014, but it is in United Nations Scientific Committee on the Effects of Atomic Radiation. UNSCEAR 1993.Report to the General Assembly, with Scientific Annexes. Why have you accepted the occupancy factor as 0.5 ? The recommended one is equal to 0.8 (UNSCEAR, 1993, IAEA, 2014), and the recommended equilibrium factor 0.4.

We checked the formula and the doses were recalculated.

Once again, we thank you for detailed feedback. We hope that questions have been properly addressed in the revised version of the article.

Sincerely,

the authors

Reviewer 2 Report

The work is interesting, well organized and documented, and uses correct grammar and syntax. The methodology and the discussion of the results are presented, with useful interpretations of the data.

General comments:

 The Authors should revise the paper.

 Specific comments:

·       The information about how representative samples were provided, what standard sampling method was used, and how the samples were stored, and transported to the laboratory is missing.

·       The methods for sample preparations should be described in detail for all proposed methods.

·       The information about equipment used for sample preparations and detections should be written and explained in detail.

·       What calibration standards were used for calibration?

·       Information on the accuracy and precision of all methods should be given and the uncertainties of measured concentrations and dose estimations should be defined (e.g. as standard deviations, standard uncertainties, expanded uncertainties with the value of k factor) and contributions to uncertainty outlined (e.g. statistical uncertainty, calibration uncertainty, sampling uncertainty).

·       A paragraph on the reliability of the measured radon concentrations by radon should be inserted.

·       The results should be written with a significant number of digits and with associated uncertainties in the text and the Tables.

Author Response

RESPONSE TO THE REVIEWER â„–2

We would like to thank the reviewer for the detailed and critical analysis of our article and constructive comments made. We agree with the comments and recommendations kindly offered by the Reviewer and these did contribute to improve the manuscript. We respond to each comment below.

-The information about how representative samples were provided, what standard sampling method was used, and how the samples were stored, and transported to the laboratory is missing.

We added information about sampling method in lines 384 to 389:

«A 2 kg duplicate sample was collected from each horizon and dried for three days at room temperature. Then, each sample was put in a plastic container, carefully sealed, and transported within 1-3 days to the laboratory in Tomsk Polytechnic University (TPU). The sampling of granite rocks, weathering crusts, and soils and the sample preparation were carried out in accordance with the Russian State Standard 17.4.1.03-83 [41]. »

Further info in lines 418-420: «Further analyses were carried out by XRD and SEM on grain-size fractions of soils (grain size fractions of 1–0.5; 0.5–0.25; 0.25–0.01; 0.01–0.04; < 0.04 mm) separated by the analytical sieve set EAH2.1, compliant with DIN ISO 3310.»

-The methods for sample preparations should be described in detail for all proposed methods.

Sample preparation was needed for INAA, and  XRD analysis. The information on the INAA sample preparation was now added in lines 413 - 417:

«The INNA method does not require chemical preparation of the sample, as it is based on the gamma spectrometric analysis of radioactive isotopes formed during the neutron bombardment of samples. For this analysis, the geological material was grinded to 100 mesh, and sample aliquots of 100 mg were wrapped in foil for neutron irradiation.»

Information on the XRD sample preparation was added in lines 447-448:

«For XRD, the test sample was crushed to powder and placed in a quartz glass cuvette.».

Information on sample preparation for f-radiography was added in lines 447-448.

-The information about equipment used for sample preparations and detections should be written and explained in detail. What calibration standards were used for calibration? A paragraph on the reliability of the measured radon concentrations by radon should be inserted.

We added now the information about equipment and calibration in the Materials and methods section,  in lines 367-370, 373-382, 387-400, 421-436.

-Information on the accuracy and precision of all methods should be given and the uncertainties of measured concentrations and dose estimations should be defined (e.g. as standard deviations, standard uncertainties, expanded uncertainties with the value of k factor) and contributions to uncertainty outlined (e.g. statistical uncertainty, calibration uncertainty, sampling uncertainty).

Information on the accuracy and precision of all methods and uncertainties of measurements was now added in lines 399-400, 381-382.

-The results should be written with a significant number of digits and with associated uncertainties in the text and the Tables.

We corrected the text and tables according to the comment.

Once again, thanks for the detailed feedback. We hope that questions have been properly addressed in the revised version of the article.

Sincerely,

the authors